# “I Was Very Shocked, I Wanted It to Be Over”: A Qualitative Exploration of Suicidal Ideation and Attempts among Women Living with HIV in Indonesia

**DOI:** 10.3390/ijerph21010009

**Published:** 2023-12-20

**Authors:** Nelsensius Klau Fauk, Gregorius Abanit Asa, Caitlan McLean, Paul Russell Ward

**Affiliations:** 1Centre for Public Health, Equity and Human Flourishing, Torrens University Australia, 88 Wakefield St., Adelaide, SA 5000, Australia; gregorius.asa@student.torrens.edu.au (G.A.A.); caitlan.mclean@torrens.edu.au (C.M.); 2Institute of Resource Governance and Social Change, Kupang 85227, Nusa Tenggara Timur, Indonesia

**Keywords:** suicidal ideation and attempt, risk factors, women, HIV, Indonesia

## Abstract

HIV diagnosis and poor HIV management have various detrimental impacts on the lives of people living with HIV (PLHIV). As a part of a large qualitative study investigating HIV risk factors and impacts, of which the topic of suicide is not a focus, this paper describes the factors contributing to suicidal ideation and attempts that arose naturally in the stories of women living with HIV (WLHIV; *n* = 52) in Yogyakarta and Belu districts, Indonesia. The participants were recruited using the snowball sampling technique. Guided by a qualitative data analysis framework, the data were thematically analysed. The findings were grouped into four main themes: (i) the women experienced immense psychological challenges due to the infection, spousal transmission, fear of mother-to-child transmission, and losing a child due to AIDS, which triggered suicidal ideation and attempts; (ii) the lack of awareness of HIV management strategies resulted in them feeling trapped and overwhelmed, and the associated negative thoughts and the anticipation and experience of HIV stigma influenced their thoughts of suicide; (iii) the lack of social support from family and friends during the early stages of HIV diagnosis, compounded with pre-existing financial difficulties, lack of income, unemployment, and feeling overburdened, also triggered the women’s thoughts of suicide; and (iv) family breakdown following HIV diagnosis, concern about future relationships, and fear of being rejected or abandoned by their partner were also influencing factors for suicidal ideation and attempts among the women. The findings indicate the need for a nuanced approach to counselling within HIV care interventions for couples to support the acceptance of each other’s HIV status whilst maintaining psychological wellbeing. Additionally, the findings indicate the importance of HIV education and awareness among community members for the de-stigmatisation of HIV and to increase the acceptance of PLHIV by their families and communities.

## 1. Introduction

The World Health Organisation has reported that suicide is a significant public health problem. Suicide is currently the fourth leading cause of death globally, with more than 700,000 deaths annually [1]. People living with HIV (PLHIV) are a group at high risk of suicide [2], with the global prevalence of suicidal behaviour reported to be higher among PLHIV than in the general population [3,4,5]. For example, the findings of a 2021 systematic review involving 185,199 PLHIV reported a high global prevalence of suicide ideation (228.3/1000 PLHIV), attempts (158.3/1000 PLHIV), and completion (10.2/1000 PLHIV) [3]. The review suggested that suicide-related deaths among PLHIV is 100-fold higher than that in the general population [3]. Similarly, another systematic review published in 2022 involving 170,234 PLHIV to assess the evidence of suicidal behaviour issues in developed and low- and middle-income countries reported suicide attempts, ideation, and completion prevalence levels of 22.3%, 9.6%, and 1.7%, respectively [5].

The factors associated with suicidal behaviour among PLHIV are varied and can be categorised into individual, psychological, and social factors. Individual-related factors include a poor quality of life, being a single parent, older age, loss of hope for a better future, loss of health insurance due to HIV infection, poor adherence to antiretroviral therapy (ART), and engagement in alcohol and substance abuse [4,5,6,7,8]. Substance abuse or drug overdosage has recently been reported in a review as the leading cause of suicide among PLHIV in the US, Europe, and Asia [2]. Additionally, low income and unemployment are individual-level risk factors associated with suicidal behaviour in PLHIV, which negatively impact the socioeconomic status of PLHIV and can result in homelessness [4,7,8,9]. Other individual-level factors for the suicidal behaviour of PLHIV include experiences of sexual and physical abuse following HIV diagnosis, which may further impact their mental health and wellbeing [6]. The mental health status of PLHIV following HIV diagnosis, including psychiatric comorbidities and disorders such as stress, depression, and anxiety, all negatively influence psychological wellbeing and are strongly associated with suicidal behaviour in PLHIV [2,5,6,7]. Depression, in particular, is reported as a common risk factor for suicidal ideation and attempts among PLHIV globally [2]. The findings of existing studies have also reported that social factors, such as being gossiped about by other people due to an HIV-positive status, a family history of suicide, living alone, lower coping self-efficacy, bullying, violence, and a lack of support from family and friends, contribute to the suicidal behaviour of PLHIV [2,4,5,9,10]. For example, the findings of several studies have reported that PLHIV with inadequate social and family support have higher rates of suicide compared to those who receive adequate support for coping with an HIV-related social–psychological burden [11,12,13].

In this study, in addition to the experiences of psychological, social, and health-related impacts of HIV, which were the focus of the investigation on [14], narratives of suicidal ideation and attempts arose naturally in the interviews with Indonesian women living with HIV (WLHIV). These indicate the significant influence of HIV on their lives following the diagnosis. Although a range of risk factors for suicidal ideation and attempts have been reported in the aforementioned studies and in current reviews [2,3,4,5], some of the naturally emerging risk factors in the stories of these Indonesian women are novel and enrich the existing findings. Over the past decade, there has been a rapid increase in HIV infection in Indonesia, from an estimated 98,390 cases in 2012 to 526,841 in September 2022 [15]. Indonesia currently carries the second largest burden of HIV in Asia [16]. Despite the high prevalence of suicidal behaviour among PLHIV reported globally and the rapid increase in and high burden of HIV in the context of Indonesia, the literature suggests that there is a paucity of evidence on suicidal behaviour among PLHIV or WLHIV in the country [15,17]. There have been some reports on suicide in the general Indonesian population, although the national data are not available [18]. For example, the WHO estimated the suicide rate to be 2.6/100,000 people [19], whereas two other surveys reported the rates of suicidal ideation to be 4.75% among adolescents and 25% among adults [20]. The paucity of evidence of suicide behaviour in PLHIV seems to reflect the lack of routine suicide assessment and the unavailability of services targeting suicide behaviour among PLHIV in the country [15]. In addition, as the existing studies globally on this topic have only used quantitative methods, with the majority employing a cross-sectional design, followed by cohort, prospective, retrospective, and longitudinal designs [2,3,4,5], there is a lack of in-depth qualitative exploration of the views and experiences of PLHIV regarding suicidal ideation or attempts. WLHIV are a highly vulnerable group carrying the double burden of their HIV-positive status. The HIV status often leads to them being perceived negatively, facing discrimination [21,22,23]. Their gender is also associated with social and cultural subordination within families and communities in developing countries [24,25,26]. Therefore, understanding their experiences of suicidal ideation and attempts and the risk factors following the HIV diagnosis is important to inform the development of policies and tailored interventions that support women in managing their negative HIV consequences. This study fills these gaps by exploring in-depth the risk factors for suicidal ideation and attempts among WLHIV in Yogyakarta and Belu districts, Indonesia. Understanding the risk factors may inform the development of evidence-based interventions targeting the needs and issues faced by WLHIV to support them in effectively coping with HIV-related challenges.

## 2. Methods

### 2.1. Study Settings

The study was conducted in Yogyakarta and Belu districts, Indonesia. Yogyakarta municipality is a district in the Special Region of Yogyakarta province. It has a total population of 636,660 people, of whom 1488 have been diagnosed with HIV, with women representing 30.6% of the total cases [27,28]. The majority of PLHIV in Yogyakarta are on ART [28]. HIV care services for PLHIV in the district include ART, HIV counselling and testing (VCT), CD4 and viral load tests, and other medical tests to support HIV treatment [29]. Yogyakarta has 20 hospitals and 27 public and sub-public health centres located within communities throughout the district [27]. HIV care services are provided in ten of the public health centres and four of the hospitals in Yogyakarta [29]. Belu is 1 of 22 districts in East Nusa Tenggara province, located in Eastern Indonesia [30]. It has a total population of 204,541 people, of whom 1200 people have been diagnosed with HIV, with women representing 39.8% of the total cases and approximately 25% of PLHIV have ever started ART [30]. ART is available in a single clinic in a small town located in the centre of the district [30]. In general, HIV care services in the district are limited to HIV counselling, testing, and ART [30]. Additionally, other medical tests, such as liver and kidney function tests, CD4 tests, and viral load tests, to support HIV treatment are not available in Belu.

### 2.2. Study Design, Participants, and Data Collection

A large-scale qualitative study that explored HIV risk factors and impacts on WLHIV was conducted in two settings in Yogyakarta and Belu, Indonesia. The data presented in this paper form part of a larger study and report on women’s experiences of suicidal ideation and attempts, which were not the focus of the initial study aim or arose naturally in the interviews with WLHIV. This study utilised a qualitative design that was considered appropriate and effective when investigating human experiences and perceptions and provided considerable insights into their real-life challenges [31,32]. The qualitative design enabled the researchers to undertake an in-depth exploration of different participants’ stories, experiences, understandings, interpretations, values, and meanings in relation to their life as well as the impact of their HIV diagnosis, and suicidal ideation and attempts [33,34,35].

This study was carried out from June to November 2019 and included 52 participants who were WLHIV, with 26 participants living in each district. The recruitment used the snowball sampling technique and started from one HIV clinic in each study setting. After soliciting permission from these clinics, the study information sheets were posted on the information boards and made available at the reception desks for distribution to clients or patients. The information sheets contained the contact details of the field researcher/first author (N.K.F.), and the potential participants who contacted (by text or phone call) and stated their willingness to participate voluntarily were recruited and assigned to a convenient time and place for the interviews.

One-to-one in-depth interviews were employed for the data collection by the field researcher. The interviews were conducted in a private room at the HIV clinic in Belu and a rented house near the HIV clinic in Yogyakarta. In addition to the audio recording of the interviews, notes were taken during the interviews, if necessary, which were then integrated into each interview during the transcription process. To enable a smooth conversation between the field researcher (N.K.F.) and each participant, all the interviews were carried out in Indonesian, which is fluently spoken by both the field researcher and participants. Only the interviewer and each participant were present in the interview room. The interviews took approximately 35 to 87 min. Regarding the topic of suicidal ideation and attempts, several main areas were explored in the interviews, which included the participants’ experience of psychological challenges; how such challenges influenced them and triggered suicidal ideation or attempts; how participants managed the HIV infection, and what strategies they used to manage the impacts of their HIV diagnosis; whether or not they experienced stigma and discrimination and how this influenced them; whether or not they received social support from others and how this influenced them; and the impact of HIV on their spousal relationships or relationships with partners. We ceased participant recruitment and interviews once we felt the data were rich enough to address our research questions and objectives and data saturation had been reached or no new themes emerged from the interviews. We regarded the similarities of responses from the last few interviewees as an indication of data saturation, which justified our decision to cease data collection [36].

### 2.3. Data Analysis

Before the comprehensive analysis of the data, audio recordings were manually transcribed verbatim, and the notes taken during the interviews were integrated into each transcript by the first author (N.K.F.). The data analysis was performed in Indonesian to maintain the sociocultural meanings attached to the information provided by the participants [37]. For this publication, the selected quotes were translated into English by N.K.F. The translation accuracy and credibility of the findings were maintained through repeated checks of the transcripts against the translated interpretations and the examination of the meaning in both languages [38]. The translation was also checked for clarity by G.A.A. and P.R.W. To support a coherent and structured way of managing these qualitative data and to enhance the rigour, transparency, and validity of the analytic process, the data analysis was guided by Ritchie and Spencer’s framework analysis for qualitative data [39]. Several steps were performed during the data analysis, including: (i) Repeatedly reading the transcripts, breaking down information in each transcript into small pieces of data extracts, and making comments or labels to them. (ii) Key issues and concepts that recurrently emerged from the information provided by the participants were written down to identify and form the thematic framework. This was an iterative process involving repeated changes and the refinement of the themes. (iii) Open coding was applied to the data extracts of each transcript, followed by close coding, through which similar or redundant codes were collated and grouped into the same themes and sub-themes. (iv) A comparison of the data or information across interviews and within each interview was performed, and then the data were examined and interpreted as a whole. Finally, (v) the mapping and interpretation of the data as whole were carried out [39,40]. Although the analysis was primarily performed by the first author, regular discussions among the team members were conducted during the analysis process and comments from team members for revisions were considered during the drafting stage. Finally, all authors agreed on the final themes and interpretation presented in this manuscript.

Ethical approvals for this study were obtained from the Social and Behavioural Research Ethics Committee, Flinders University (No. 8286), and the Health Research Ethics Committee, Duta Wacana Christian University (No. 1005/C.16/FK/2019). Before commencing the interviews, participants were again informed about the purpose of the study and the voluntary nature of their participation, which meant that they had the right to cancel their participation in the study at any time without consequence. Consent forms were signed by each participant and returned to the field researcher on the interview day. After the interviews, each participant received IDR 100,000 (approximately USD 7) of reimbursement for their time and transport.

## 3. Results

### 3.1. Sociodemographic Profile of the Study Participants

The WLHIV in this study were aged between 18 and 60 years, with the majority being between 20 and 39 years old (*n* = 34) (see Table 1). Nearly half of them were married or remarried (*n* = 23), while the rest never married (*n* = 8), were divorced (*n* = 6), or were widowed (*n* = 15). Participants had been living with HIV for different time periods, with the majority between one and five years (*n* = 34). Several women also reported having previously been diagnosed with herpes, candidiasis, or tuberculosis. All participants were on ART when the study was conducted. Just over half of the participants reported being engaged in a range of professions, while the rest reported being either full-time housewives or unemployed.

The findings were grouped into four main themes: (i) psychological challenges and suicidal ideation and attempts among WLHIV; (ii) lack of awareness of HIV management strategies and anticipated and experienced stigma; (iii) lack of support during the early stages of HIV diagnosis and an overloaded burden; and (iv) broken families and relationships and suicidal ideation and attempts. The elaboration of these themes is presented below.

### 3.2. Psychological Challenges and Suicidal Ideation and Attempts among WLHIV

Being diagnosed and then living with HIV were not easy for these women considering the negative social, cultural, and religious perceptions attached to HIV status. Thus, the challenges that they faced were not only related to HIV, care, treatment, and management (i.e., how to access ART, manage ART adherence and side effects, afford medical and transport costs, etc.), but also how to prevent negative social consequences, such as stigma and discrimination, which are still prevalent in the context of the study settings. A tremendous psychological pressure was placed on the participants to manage different aspects of their lives following HIV diagnosis, whilst at the same time, they had to face worries and thoughts about the various possible negative impacts that may lie ahead in the face of their diagnosis. Such pressures were triggered by the fear of transmitting the virus to their children, concerns about the negative impacts of their condition on their future and the future of their children, the possibility of causing shame to their families, feeling guilty towards their families, self-blame and blaming of their husband, and a fear of being rejected or left by their partner:


*“At that time (once she was diagnosed with HIV), I was shocked and very much stressed out. …. I cried every day because I was stressed out thinking about this (HIV). I felt like my future was already blurred. I felt broken. If I remember that moment, I still cry.”*

*(Married, Yogyakarta)*



*“Sometimes I cry because I have been thinking of my children: one is still little and another is still in my womb (7 months). What would happen if they get (HIV)? Hope I bear this burden myself.”*

*(Remarried, Belu)*



*“I am often stressed out because of this condition (having HIV-positive status) and fear whether there will be a man who accepts and wants to have a relationship with me. What I am thinking about is that there would not be a man who wants to marry a woman with HIV like me. This makes me scared of my future.”*

*(Never married, Belu)*


The psychological pressures experienced by the participants were often exemplified through feelings of stress, depression, hopelessness, and feeling broken, which at a certain point made some of them consider or attempt to end their lives. For the women who were mothers, the psychological stress seemed to be exacerbated by the fact that they had transmitted HIV to their children or that their children had died of AIDS. Mother-to-child-transmission and the loss of a child due to AIDS often caused self-blame, guilt, increased psychological burden, and triggered suicidal ideation among them:


*“I was broken and desperate after the diagnosis. I was very depressed. I felt like the world was dark and often the thought of ending my life came across my mind very often. It took a long time to recover. It is getting better, but sometimes I am still desperate until now (ART).”*

*(Widowed, Yogyakarta)*



*“After my son passed away (died from AIDS) and I tested positive for HIV (her son was first diagnosed with HIV once he was critically ill), I locked myself in my room for a while and during that time I sometimes thought of ending my life because the burden I felt was so heavy.”*

*(Widowed, Belu)*


HIV transmission through a spouse or boyfriend, whether intentional or not, was another factor that greatly influenced the psychological condition of the participants and how they reacted towards their HIV-positive status. Some women shared reactions, including feeling anger and the unacceptance of their HIV status, as well as reporting attempted suicide or that they planned to harm themselves and their spouse:


*“I am furious because my husband gave me HIV. I am 60 years old and a former teacher; what would people say if they discovered that I have HIV? My husband went to Bali, slept with prostitutes, and contracted HIV. He passed it on to me. I am furious and one time, I prepared poison at night to put in his drink and mine in the morning so that we would die together. …”*

*(FP, married, Belu)*



*“When I was told (by her ex-boyfriend) that I must have contracted HIV, I felt broken, stressed out. I felt like I could not breathe anymore. I told him, ‘you destroy my life’. I was broken, and wanted to commit suicide. …. I was very shocked, I wanted it to be over. I took a lot of medicines (pills), I bought many types of medicine and took them at once.”*

*(Never married, Yogyakarta)*


### 3.3. Lack of Awareness of HIV Management Strategies and Anticipated and Experienced Stigma

Employing strategies for HIV management following diagnosis is critical for PLHIV to manage the negative consequences of living with HIV. The stories of some participants portrayed a lack of awareness of HIV management strategies. This was reflected in the participants not knowing what to do or what solutions needed to be undertaken to address HIV-related pressure and the burdens they faced due to their diagnosis. For some women, such a situation made them feel trapped and overwhelmed with their own negative thoughts and this led them to consider ending their lives. Similarly, having no one to talk to and thinking about the possible consequences that could occur due to the infection were also reported as triggering the participants’ thoughts of suicide:


*“The thought of ending my life came across my mind every time I thought about a range of consequences to face later in life due to this infection. I felt overwhelmed by all those thoughts. It was very difficult moments, sometimes I held the knife (to commit suicide) but then threw it away because I felt scared of doing that (suicide).”*

*(Married, Yogyakarta)*



*“I was struggling alone after I was diagnosed with HIV. I had nobody who I really trusted to talk to. I was on my own. I could not tell my family (parents and siblings) or close friends because I was scared. It felt like I was stuck, making me think it is better to die, just die, and the problem will be gone. I thought about that (ending her life), but I did not try to do it. Now, I live with my brother, and he fully supports me.”*

*(Never married, Belu)*


These negative thoughts, including the potential negative social consequences they may face, appeared to lead to a strong anticipation of stigma and influenced the participants’ support-seeking behaviours. Some women described being embarrassed and afraid to talk openly about what they felt and experienced to their family members for fear of negative reactions. Likewise, the fear of spreading information about their HIV-positive status, which could lead to further negative consequences and worsen their health and psychological conditions, prevented some women from being open or seeking support from friends or significant others. Fear was particularly apparent during the early stages of their HIV diagnosis or infection. The stigma anticipated and experienced after HIV diagnosis added additional pressure and triggered suicidal thoughts among some participants:


*“Sometimes, all negative thoughts about any possible consequences that could happen to me next made me think it is better to end my life. I am embarrassed to discuss my condition with family members and friends. I am afraid they will ask many questions I may not be able to answer openly.”*

*(Never married, Yogyakarta)*



*“As I said before, my mom rejected me for months once she knew that I have HIV. I felt like I did not want to live anymore; I just wanted to finish my life.”*

*(Divorced, Yogyakarta)*


The stories of some participants in rural Belu also showed that the stigma they anticipated and experienced had led to self-isolation, which was also a contributor to suicidal ideation:


*“So far, I avoid other people. I do not talk to my neighbours because I am afraid they will know (about her HIV status). …. I spent most of my time in the house in my room. It is difficult, and sometimes I think perhaps death is a good way to end this suffering.”*

*(Remarried, Belu)*


### 3.4. Lack of Support during the Early Stages of HIV Diagnosis and an Overloaded Burden

Support from family, friends, and other people, whether in the form of emotional, informational, or material support, is a significant factor that provides psychological protection for PLHIV. It also supports them in dealing with the various forms of HIV-related challenges they may face following the diagnosis. The interviews showed that some women in this study did not receive support from family or friends who knew about their HIV-positive status. This not only made dealing with their poor physical health conditions more difficult, but also triggered suicidal ideation:


*“My mom did not support me. She was mad at me, ignored me, and blamed me for everything. She thought this was my fault. I was on my own for several months, and it was so difficult. I was locked in my room by myself. There were times I thought about suicide to end my life. My daughter was the reason I did not do it. …”*

*(Divorced, Yogyakarta)*


Being diagnosed with HIV exacerbated a range of existing burdens placed on WLHIV and their families. The difficulties they experienced before being diagnosed with HIV, such as economic or financial difficulties, lack of income, unemployment, and the inability to meet the needs of children, were often exacerbated by an HIV diagnosis and became an overloaded burden for WLHIV and their families. Overloaded burden, teamed with a lack of solutions and coping strategies, often worsened their psychological condition and triggered thoughts of ending their life or suicide. Such experiences are reflected in the stories of the following participants:


*“My family’s situation is quite difficult. We (the woman and her husband) often have difficulty meeting our daily needs and the school needs of our children. Unfortunately, I fell ill and was diagnosed with HIV. This is what makes me think negatively about our lives, including thinking about ending my life and all the suffering. I know that my husband transmitted it to me, and he was also diagnosed with HIV at the same time as me because we were both sick and admitted to the hospital. Our family life was already quite difficult, and now it is even more difficult because of this [HIV infection]. …”*

*(Married, Belu)*



*“If I didn’t have my daughter, I might have committed suicide. My husband died of HIV. He transmitted HIV to me, and I transmitted it to my daughter. The burden I am carrying is heavy. I don’t have a job, and our life depends entirely on the support of my parents. …”*

*(Widowed, Yogyakarta)*


### 3.5. Broken Families and Relationships and Suicidal Ideation and Attempts

Being infected with HIV was associated with a negative impact on the women’s relationships within their families. Family disputes and husband–wife separation or divorce were examples of the impact on spousal relationships that occurred due to the women’s HIV diagnosis or status. Several women in both study settings mentioned that family disputes and husband–wife separation or divorce occurred because of the unacceptance of their HIV status by their family or husband. The participants faced accusations from their extended families and husbands regarding who first transmitted the virus:


*“The family of my husband said (to her family) that I was not a good woman (after her child died from AIDS), my family said (to her husband’s family) ‘your son is not a good man’. They blamed each other. …. After my child passed away, my husband’s parents separated him from me; it was so stressful. …. Initially, the separation from my husband felt very difficult because I had just lost our child. On one day, I nearly took baygon [an insecticide used for extermination of mosquitoes], but my mom saw it and threw it away.”*

*(Widowed, Belu)*


Mother–child separation was another negative impact of HIV on the women’s family relationships following their HIV diagnosis. Several women interviewed in Yogyakarta described how their children were forcibly separated from them by their mothers and in-laws because of the fear of transmitting the infection to their children. This not only increased their psychological burden but also triggered suicidal ideation in some participants, as reflected in the following narrative:


*“My child was kept away from me (by her mother). She said ‘do not get close to your child, you could transmit the virus to your child’. I was not allowed to touch my child for three months. It was very painful and stressful. I could see my daughter but was not allowed to hug her. This was also one of the things that made me think of ending my life.”*

*(Divorced, Yogyakarta)*


Concerns about future relationships and fear of being rejected or left by their future partners due to their HIV-positive status also increased the psychological challenges faced by women who had not been married in both study settings. Such concerns were associated with adverse psychological effects, such as desperation, sadness, worry, stress, and suicidal ideation due to feeling unworthy. These effects were underpinned by the women’s previous experiences or knowledge of other PLHIV’s experiences of being left by someone they loved because of their HIV-positive status:


*“There was a guy who was close to me (in a relationship with her). We were serious in the relationship, but once he knew about my (HIV) status, he avoided me step by step, and finally we lost contact. …. I was so desperate for several months. During those months, sometimes I thought it was probably better if I die because I felt like I was not worthy.…”*

*(Never married, Yogyakarta)*



*“I am often stressed out because of this condition (having HIV-positive status) and fear whether there will be a man who accepts and wants to have a relationship with me. What I am thinking about is that there would not be a man who wants to marry a woman with HIV like me. This makes me scared of my future …. The thoughts about suicide have often crossed my mind. One of the reasons is that I keep thinking about my future. I want to get married, but it feels impossible because I have HIV. I have often heard rejection of people who have HIV.”*

*(Never married, Belu)*


## 4. Discussion

HIV diagnosis and poor HIV management often have detrimental impacts on PLHIV in many aspects. Although suicidal ideation and attempts were not the focus of the initial design of the study, the topic emerged in the participants’ stories as part of their lived experiences following HIV diagnosis and management, reflecting the significant influence of HIV on their lives. Not dissimilar to previous findings on suicidal issues among PLHIV in other settings [2,5,6,7], the current findings emphasise the tremendous psychological challenges faced by WLHIV. The challenges stemmed from the negative social, cultural, and religious perceptions associated with HIV; the struggle with and concerns of a possible HIV transmission to their children; and negative impacts on the future and their family’s reputation. These challenges often lead to self-blame and guilt, further intensifying the emotional turmoil and contributing to thoughts of suicide among the participants. The findings indicate the immense burden that HIV presents in these women’s lives, which extends beyond medical challenges, and reveal the importance of comprehensive HIV interventions that address the mental health needs of these women. For example, psychological support interventions that address the HIV-related depression, stress, and anxiety faced by WLHIV, which have been reported as being effective in other settings [41,42,43], can be beneficial to improve the mental health and wellbeing of the women in this study. Such interventions are critical, especially for WLHIV in the rural communities in Belu district, where healthcare and mental health support for PLHIV are lacking or are very limited [14,44,45].

The study suggests the importance of individual knowledge of HIV management strategies, which can help WLHIV to respond positively to the infection and empower them to navigate their lives following the diagnosis. Such support and increased awareness may improve their access to counselling, the available social support, HIV care, and mental health services [14,46,47,48]. The lack of knowledge of HIV management strategies among the participants led them to feeling overwhelmed by negative thoughts, which ultimately increased their risk of suicidal ideation. HIV stigma and discrimination among the participants’ families, communities, and within healthcare settings are also factors that influence suicidal ideation. Such experiences exacerbated the women’s psychological and social conditions, leading to strong anticipated stigma, self-isolation, and suicidal ideation. The current findings enrich those of previous studies reporting how HIV stigma and discrimination have negatively influenced the social life of PLHIV, their access to HIV care and treatment, education, and employment [22,49,50]. Thus, the current study indicates the importance of social networks for WLHIV to support them in mitigating psychological distress, managing the infection, and navigating their lives. The social networks for and among WLHIV could be established in collaboration with the healthcare system or in the form of peer support groups in the study settings, which have been reported as being effective in supporting PLHIV to navigate their HIV treatment and social interactions [51,52]. The women’s lack of social networks and support led to them feeling overwhelmed and isolated. In the absence of emotional, informational, and material support, the burden of their HIV diagnosis and management placed on the participants was exacerbated and the risk of suicidal ideation increased.

For the participants in this study and many other PLHIV in different settings globally, such disadvantaged conditions exacerbate pre-existing socioeconomic factors, including unemployment and financial instability, which results in PLHIV facing more impoverished living conditions [53,54]. These results support the findings of previous studies suggesting that poor socioeconomic conditions, reflected in low income, unemployment, financial difficulties, and poor living conditions, are contributors to suicidal behaviours among PLHIV [4,7,8,9]. The findings have important implications for HIV interventions in the study settings to emphasise the critical role of social networks for PLHIV and foster supportive environments and the importance of destigmatising HIV [55,56,57]. This is in line with previous findings suggesting that social support interventions for WLHIV were effective in improving their self-esteem, the social support they received, as well as their social and emotional wellbeing [41].

The study also reports novel findings on the dynamics of WLHIV within spousal relationships, which are influenced by HIV transmission through spouses or husbands. The participants showed a rejection of being infected by spouses, and this seemed to be the underlying factor influencing their reduced psychological wellbeing. Relationship dynamics in the face of an HIV diagnosis triggered adverse emotional reactions from the participants, such as anger, stress, and resentment, and subsequently, these triggered suicidal ideations and attempts on their own lives as well as those of their spouses. This situation also reflects how difficult it is for these women to deal with the experience of being diagnosed and living with HIV in the context of their spousal relationships. The study suggests that suicidal ideation and attempts among WLHIV are also triggered by negative reactions from their spouses and families relating to their HIV diagnosis. This has not been reported in the previous findings synthesised in global reviews supporting factors for suicidal ideation and attempts [2,3,4,5]. The negative reactions were reflected in spousal disputes, separation, and divorce, often driven by the unacceptance of the women’s HIV status and mother–child separation due to fear of transmission. The findings indicate the profound impact of HIV on family dynamics, which often exacerbates the women’s psychological distress and burden and triggers suicidal ideation. Covering the experiences of mothers or married women living with HIV who brought up specific stories of HIV-related challenges, such as losing custody of their children, the burden of transmitting HIV to their child and their child having died from AIDS, and rejection by their spouse, the current study adds aspects of importance that would have been missed if focusing on men or other groups of PLHIV.

Our findings also suggest that factors such as concerns about future relationships and fear and the experience of rejection in relationships profoundly influenced the women psychologically and triggered suicidal ideation. This was especially apparent among the unmarried women in this study. Previous studies have suggested that rejection in relationships experienced by PLHIV is mainly due to the fear of HIV transmission, stemming from a lack of knowledge and the negative perception associated with HIV [14,49,50]. The findings have important implications for the adoption of a nuanced approach to counselling within HIV care interventions or programs for seroconcordant and serodiscordant couples to support the acceptance of each other’s HIV status and psychological wellbeing [58,59]. Similarly, the findings indicate the importance of raising HIV awareness among community members to increase the acceptance of PLHIV within families and communities and the social and mental health wellbeing of PLHIV [60,61].

### Limitations and Strengths of the Study

The current findings should be interpreted with caution as they present some limitations. Using the snowball sampling technique, the participants in this study were recruited through networks of participants who were already on ART. Thus, it is highly likely that we under-sampled WLHIV outside the networks of the current participants. We also acknowledge that we did not include WLHIV who were not on ART and may have different experience of HIV diagnosis and management from the current participants. Therefore, the present findings reflect the specific experiences of WLHIV in the study settings, which may be dissimilar to those of other WLHIV in other settings. Thus, as is the case of many qualitative studies, the current study cannot be generalised to all WLHIV living with HIV in Indonesia and globally. Another possible limitation is that, as the study focuses on exploring HIV-related risk factors for suicidal ideation and attempts, this might have led us to overlook other potential risk factors that are not directly related to HIV. However, this is the first qualitative study that explored suicidal ideation and attempts in-depth among Indonesian WLHIV. Thus, the findings are essential to inform HIV policy and interventions to address the challenges faced by and the needs of Indonesian WLHIV and their families.

## 5. Conclusions

This study presents the multifaceted challenges faced by WLHIV, encompassing psychological and social dimensions at the individual and family levels, which contribute to suicidal ideation and attempts. The women experienced immense psychological challenges, such as stress, depression, hopelessness, and feeling broken due to the infection, spousal transmission, fear of mother-to-child transmission, and losing a child due to AIDS, which triggered suicidal ideation and attempts. Other risk factors for suicidal ideation and attempts among these women included a lack of awareness of HIV management strategies, the associated negative thoughts and the anticipation and experience of stigma due to their HIV status, and a lack of social support from family and friends, compounded with pre-existing financial difficulties, lack of income, unemployment, and feeling overburdened. Family breakdown following HIV diagnosis, reflected in family disputes, husband–wife separation or divorce, mother–child separation, concerns about future relationships, and fear of being rejected or abandoned by their partner due to their HIV positive status, were also influencing factors of suicidal ideation and attempts among the women. The findings indicate that, in order to address these challenges and reduce the risk of suicidal ideation and attempts among WLHIV, there is a need for comprehensive and holistic support programs, which may include mental health support, HIV education and awareness campaigns for the enhancement of community knowledge, stigma reduction and acceptance of PLHIV, and interventions to strengthen support networks for WLHIV and their families. Perinatal interventions around pregnancy, birth, and mothering are also recommended as these can help WLHIV to take positive responses to overcome the fear of and prevent mother-to-child-transmission and support them in raising their children. Future large-scale studies addressing comprehensive factors that contribute to suicidal ideation and attempts among WLHIV in different settings in Indonesia are recommended as the results may inform the better development of HIV policies and practices in the country.

## Figures and Tables

**Table 1 ijerph-21-00009-t001:** Sociodemographic profile of the study participants.

Characteristics	Women Living with HIV	
	Yogyakarta (*n* = 26)	Belu (*n* = 26)
**Age**		
18–19		2
20–29	6	4
30–39	12	12
40–49	8	6
50–59		1
60–69		1
**Marital status**		
Single/never married	5	3
Divorced	5	1
Widowed	3	12
(Re)Married	13	10
**HIV diagnosis**		
1–5 years ago	16	18
6–10 years ago	7	7
11–15 years ago	3	1
**Other infections**		
Herpes	2	
Candidiasis	1	3
TB	4	5
**Education**		
University graduate/Diploma	6	6
Senior High school graduate	13	5
Junior High school graduate	6	6
Elementary school graduate	1	8
Elementary school dropout		1
**Occupation**		
Housewife	11	13
Entrepreneur	3	6
Tailor, Laundress, Shopkeeper	3	
Sex worker	1	
NGO worker	3	
Nurse/health worker	1	2
Private employee	2	2
Banker	1	
Retired civil servant		1
Civil servant		1
University student	1	

## Data Availability

The data presented in this study are available upon request from the corresponding author. The data are not publicly available due to restrictions set by the human research ethics committee.

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
