# Peer review of "“I Was Very Shocked, I Wanted It to Be Over”: A Qualitative Exploration of Suicidal Ideation and Attempts among Women Living with HIV in Indonesia"

_ijerph, 2023, doi:10.3390/ijerph21010009_

Round 1

Reviewer 1 Report

Comments and Suggestions for Authors

Dear Editor,

Thank you for the opportunity to review the manuscript titled: “I was very shocked, I wanted it to be over”: A qualitative exploration of suicidal ideation and attempts among women living with HIV in Indonesia. Herewith my feedback:

Introduction:

The authors have done well in contextualizing the study and discussed several systematic reviews to underscore the issue of suicide among people living with HIV/AIDS. In addition, they have detailed some of the factors that contribute to suicidality among this group. They have also touched on the mental health impact of living with HIV/AIDS. The authors have also discussed the prevalence rates for HIV in Indonesia and highlighted the gap that their study aims to address. Overall, this is a succinct and well written portion of the manuscript.

An area that the authors could potentially expand on is the differential vulnerability of women to contracting HIV and the gendered/patriarchal ideologies and socio-economic circumstances that may contribute to women’s increased vulnerability, particularly within the Indonesian setting. The authors may also want to consider including a theoretical framework that can serve as the lens for their study, as this is the norm in qualitative research.

Methodology:

The study setting, research design and data analysis sections are well written and provide comprehensive insights into the methodology for the study. It is also evident that the researchers promoted the rigor of their research throughout the process and used appropriate techniques to achieve this.

Results:

The authors have provided a comprehensive demographic profile of the participants. I would suggest that prior to 3.2, the authors state that they are presenting the major themes that emerged from participants narratives. This will help to improve the flow of this section. I would also recommend that the authors consider adding a table of the major themes.  The quotes from participants are reflective of their lived experience and aligned with the theme.

Discussion

I would recommend that the authors expand on their recommendations for intervention, grounded in the distinctive findings of their study. The findings could also be framed within an appropriate theoretical framework.

Comments on the Quality of English Language

Only minor editing needed. 

Author Response

Dear Reviewer,

Thank you very much for your valuable comments for the improvement of our manuscript. We do appreciate your time and contribution. Our responses are attached.

Kind regards,

Fauk

Reviewer 2 Report

Comments and Suggestions for Authors

INTRODUCTION

1. The study focuses specifically on women living with HIV (WLHIV) in Yogyakarta and Belu districts, Indonesia. While this provides in-depth insights into this group, the findings may not be generalizable to all PLHIV populations, especially in different cultural or geographical contexts.

2. The introduction heavily references global statistics and studies on PLHIV and suicidal behavior. However, it may lack a direct comparative analysis between these global trends and the specific situation in Indonesia, particularly in the Yogyakarta and Belu districts.

3. The introduction mentions that previous studies have primarily used quantitative methods. While this study aims to address this gap with qualitative analysis, it's important to note that qualitative methods have their limitations, such as potential biases in narrative interpretation and challenges in quantifying the prevalence of suicidal ideation and attempts.

4. The introduction lists a wide range of risk factors for suicidal behavior among PLHIV. However, it may not clearly delineate which of these factors are most prevalent or significant in the Indonesian context, particularly for WLHIV in the study areas.

5. There is a mention of the paucity of evidence on suicidal behavior among PLHIV in Indonesia. This gap in national data could limit the ability to fully understand the context or to compare it with global trends.

6. While the study aims to explore HIV-related risk factors for suicidal ideation and attempts, it might overlook other potential contributing factors that are not directly related to HIV but could be significant in the lives of WLHIV.

7. The introduction suggests that some emerging risk factors in the stories of these Indonesian women are novel. This focus on novelty might overshadow the importance of well-established risk factors that are crucial for understanding the broader context of suicidal behavior among WLHIV.

8. While the study aims to inform the development of interventions, the introduction does not provide a clear framework or direction for how these interventions might be designed or implemented based on the study’s findings.

9. The introduction could benefit from a more detailed exploration of how cultural and socioeconomic factors specific to Indonesia influence suicidal behavior among WLHIV, beyond the global context.

10. While social factors are mentioned, there might be a need for a more in-depth discussion on how stigma and discrimination specifically related to HIV impact suicidal behavior among WLHIV in Indonesia.

METHODS

1. The use of snowball sampling, while useful for reaching a specific population, may introduce selection bias. Participants in such samples may share similar characteristics or viewpoints, which might not represent the broader population of WLHIV in Indonesia.

2. While in-depth interviews can provide rich qualitative data, relying solely on self-reported experiences may lead to subjective biases. Participants might withhold information or alter their stories due to social desirability or recall bias.

3. The interviews were conducted in Indonesian and later translated for publication. This raises concerns about the potential loss of nuanced meanings during translation, which could affect the interpretation of the data.

4. The determination of data saturation based on the similarity of responses from the last few interviewees might be subjective. There's a risk that important variations or unique experiences might be overlooked.

5. Offering reimbursement for participation could potentially influence the decision of individuals to participate, which might affect the sample's representativeness.

6. The use of Ritchie and Spencer’s framework analysis is a structured approach, but it might constrain the emergence of unanticipated themes or insights, as it relies on predefined themes and codes.

7. The wide range in interview lengths (35 to 87 minutes) could indicate variability in the depth and detail of information gathered from each participant.

8. The focus on WLHIV’s experiences of suicidal ideation and attempts might not fully capture the broader context of their lives, including other significant factors influencing their mental health.

RESULTS

1. While the study includes participants aged 18 to 60, the majority are between 20 and 39 years. This may limit insights into the experiences of older WLHIV, who might face different challenges and perspectives on suicidal ideation and attempts.

2. The study includes a diverse range of marital statuses, but it's unclear how these different statuses might uniquely impact the experiences of suicidal ideation and attempts. More nuanced analysis based on marital status could provide deeper insights.

3. The majority of participants have been living with HIV for one to five years. This could limit understanding of the long-term psychological impacts of living with HIV on suicidal ideation and attempts.

4. Some participants reported being diagnosed with other conditions like herpes, candidiasis, or tuberculosis. The study does not delve into how these comorbidities might compound the psychological challenges and influence suicidal ideation.

5. All participants were on ART, but the study does not explore how adherence to ART, or the lack thereof, might relate to their psychological challenges and suicidal ideation.

6. The study mentions employment status but does not deeply explore how being employed, unemployed, or a full-time housewife impacts the psychological well-being and suicidal tendencies of WLHIV.

7.  While the study identifies psychological challenges like stress, depression, and hopelessness, it could benefit from a more in-depth analysis of these psychological states and their direct correlation with suicidal ideation and attempts.

8.  The study highlights stigma and discrimination as significant factors but does not provide a detailed analysis of how these experiences specifically contribute to suicidal ideation and attempts.

9. The lack of support is mentioned as a factor contributing to suicidal ideation, but the study could further explore the nature and effectiveness of different types of support (emotional, informational, material) in mitigating these thoughts.

10. The impact of HIV on family relationships is discussed, but the study could delve deeper into how these dynamics specifically lead to suicidal ideation, beyond the general mention of disputes and separation.

11. The study relies on narratives to present its findings, which could lead to selective reporting based on the most compelling or common stories, potentially overlooking less common but equally significant experiences.

12. While the study is set in Indonesia, there could be more emphasis on how the specific cultural and social context of the country influences the experiences of WLHIV and their suicidal tendencies.

DISCUSSION

1. The participants were recruited through networks of those already on ART, which could lead to a selection bias. This approach may under-sample WLHIV who are not part of these networks or those not on ART, potentially missing diverse experiences and perspectives.

2.  The study primarily focuses on the experiences of mothers or married women, which provides valuable insights but may overlook the unique challenges faced by unmarried women, widows, or those without children.

3.  While the study discusses the impact of HIV on family dynamics, it could benefit from a more detailed exploration of how these dynamics specifically contribute to suicidal ideation, beyond general mentions of disputes and separation.

4.  The study is set in Indonesia, and while it acknowledges the influence of social, cultural, and religious perceptions, a deeper analysis of how these specific cultural factors impact WLHIV's experiences with suicidal ideation could enhance understanding.

5. The discussion touches on socioeconomic factors like unemployment and financial instability, but a more detailed analysis of how these factors specifically contribute to suicidal ideation among WLHIV would be beneficial.

6.  The study suggests the importance of interventions and support networks but could provide more specific recommendations or strategies for implementing these supports effectively in the context of the study settings.

7. While the study discusses stigma and discrimination, it could delve deeper into the mechanisms by which these experiences directly lead to suicidal ideation and attempts, providing a more nuanced understanding.

8.  The study highlights the need for comprehensive mental health interventions but could further explore the current gaps in mental health support for WLHIV in the study areas and propose specific solutions.

9.  The study claims to report novel findings, particularly regarding spousal relationships and HIV transmission dynamics. However, it could benefit from a more detailed comparison with existing literature to highlight what aspects are truly novel and what implications these have for future research and interventions.

10. The discussion acknowledges some limitations but could benefit from a more thorough reflection on the methodological choices and their impact on the findings, such as the choice of qualitative design, interview methods, and data analysis techniques.

11. While the study aims to inform policy and interventions, a more detailed discussion on how these findings can be translated into practical, actionable strategies for HIV policy and interventions in Indonesia would strengthen its impact.

CONCLUSION

1. While the study recommends comprehensive and holistic support programs, it could provide more specific guidance on how these programs should be designed and implemented, especially in the context of the unique challenges faced by WLHIV in Indonesia.

2. The conclusions are drawn from a study conducted in two specific districts in Indonesia, which may limit the applicability of the findings and recommendations to other regions or populations with different cultural, social, or economic contexts.

3. The emphasis on perinatal interventions, while important, might overlook other critical stages in the lives of WLHIV, such as adolescence, menopause, or aging, which could also significantly impact their mental health and risk of suicidal ideation.

4. The study suggests interventions for stigma reduction and support network strengthening but could benefit from a deeper exploration of the mechanisms by which these interventions could effectively reduce suicidal ideation and attempts.

5. The conclusions mention the need for comprehensive support but could further emphasize the role of socioeconomic factors, such as poverty, unemployment, and education, in contributing to the challenges faced by WLHIV.

6. While the study acknowledges the need for HIV education and awareness campaigns, it could highlight the importance of culturally sensitive approaches that are tailored to the specific beliefs, values, and practices of the Indonesian context.

7. The conclusions could benefit from a reflection on the study's methodology, including its strengths and limitations, and how these might impact the interpretation and application of the findings.

8. The study aims to inform HIV policy and practice, but the conclusions could provide more detailed suggestions on how policymakers and practitioners can use these findings to develop targeted interventions and policies.

9. While the study calls for future large-scale studies, it could specify particular areas or aspects that require further investigation, such as longitudinal studies to understand the long-term impact of HIV on WLHIV's mental health.

10. The conclusions could discuss how the recommended interventions and support programs can be integrated with existing health and social services to ensure a coordinated and efficient approach to supporting WLHIV.

11. The study could emphasize the importance of engaging various stakeholders, including WLHIV, healthcare providers, policymakers, and community leaders, in the development and implementation of the recommended interventions.

12. The conclusions could suggest the need for robust monitoring and evaluation mechanisms to assess the effectiveness of the recommended interventions and support programs, ensuring they meet the needs of WLHIV and lead to tangible improvements in their mental health and quality of life.

Author Response

(The authors gave the same response as above.)

Reviewer 3 Report

Comments and Suggestions for Authors

Τhe choice of the subject is very good and interesting, although the methodology needs to be better explained. For example, the subthemes that leads to themes have not been described. The plagiarism report is attached for additional modifications (43%).

Abstract: The abstract should be a total of about 200 words maximum. The number of participants should also be included.

Study design, participants, and data collection: There ought to be more details provided about data gathering. When they took place (time period), by whom, in which place, who was present.

Data analysis:

-        The process used to identify the themes and subthemes, as well as who carried out these actions, need to be more clearly explained.

-        The participants' signed consent forms included their consent for audio recording and publication of the article, which included their statements? Results: The subthemes have not been described; it is proposed to add a table indicating the themes and subthemes so that the choice of specific themes is clear. It is crucial to discuss these facts.

Discussion: It is important to provide more specific recommendations rather than merely a generalization regarding the practical applicability of the study results. What recommendations do you have for design of relevant interventions, to incorporate the results of your study?

Conclusion: The implication of results and future research directions should be included in the Discussion section. Conclusions must be drawn from the results of the study in question, i.e. they must be condensed and interpreted in a relevant manner.

References: See the instructions to authors for the correct formatting of the bibliography.

Comments on the Quality of English Language

Minor editing of English language required

Author Response

(The authors gave the same response as above.)

Reviewer 4 Report

Comments and Suggestions for Authors

 1.      The abstract is very confusing since I cannot see the results easily. The main findings need to be presented clearly here in a more concise way. For example, how many themes have you identified?

2.      The introduction failed to describe the research gap. I could not see the clear reason for doing this qualitative study, though you said most studies used quantitative data.

3.      The main themes identified in this study were not clear. I can see they might be in the sections from 3.2 to 3.5. However, the authors need to describe how many themes identified in this study at the beginning of the results chapter.

4.      The findings of this study were not interesting though the topic is important. No surprising or unexpected theme was found. The results can be expected without this study. The implication of this study is limited.  

Author Response

(The authors gave the same response as above.)

Round 2

Reviewer 2 Report

Comments and Suggestions for Authors

Dear Author

Thanks for addressing the comment. Please carry forward the research further. Best wishes.

Reviewer

Author Response

Dear Reviewer,

Again, thanks for your previous comments for the improvement of our manuscript.

Kind regards.

Reviewer 3 Report

Comments and Suggestions for Authors

Dear authors,

You have addressed all the suggested corrections to the manuscript. It would be better to move the title 3. Results and subtitle 3.1 to the next page, and the table should be on one page only.

Author Response

Dear Reviewer,

Again, thanks for your previous comments for the improvement of our manuscript. We have address additional comment you raised.

Kind regards.

Reviewer 4 Report

Comments and Suggestions for Authors

The revised version reads much better than the previous one. It is now ready for publication in this journal. 

Author Response

(The authors gave the same response as above.)
